# Species and Strain Variability among *Sarcina* Isolates from Diverse Mammalian Hosts

**DOI:** 10.3390/ani13091529

**Published:** 2023-05-03

**Authors:** Marie Makovska, Jiri Killer, Nikol Modrackova, Eugenio Ingribelli, Ahmad Amin, Eva Vlkova, Petra Bolechova, Vera Neuzil-Bunesova

**Affiliations:** 1Department of Microbiology, Nutrition and Dietetics, Faculty of Agrobiology, Food and Natural Resources, University of Life Sciences Prague, 165 00 Prague, Czech Republic; makovska@af.czu.cz (M.M.); killer@iapg.cas.cz (J.K.); modrackova@af.czu.cz (N.M.); ingribelli@af.czu.cz (E.I.); ahmadamin@af.czu.cz (A.A.); vlkova@af.czu.cz (E.V.); 2Laboratory of Anaerobic Microbiology, Institute of Animal Physiology and Genetics, Czech Academy of Sciences, 142 20 Prague, Czech Republic; 3Department of Ethology and Companion Animal Science, Faculty of Agrobiology, Food, and Natural Resources, University of Life Sciences Prague, 165 00 Prague, Czech Republic; bolechova@af.czu.cz

**Keywords:** animals, mammalians, microbiota, *Sarcina* spp., cultivation, taxonomy

## Abstract

**Simple Summary:**

Sporadic but repeated occurrences of *Sarcina* spp. indicate that these microorganisms with atypical morphology forming packets in the fecal microbiota of animals without health problems may not always be pathogenic and seem to be a common part of the gut microbiota of various mammals. Aside from that, genotyping characterization indicates species and strain variability among *Sarcina* isolates and the potential presence of two novel taxonomic units originating from dog and elephant hosts.

**Abstract:**

*Sarcina* spp. has been isolated from the gastrointestinal tracts of diverse mammalian hosts. Their presence is often associated with host health complications, as is evident from many previously published medical case reports. However, only a handful of studies have made proper identification. Most other identifications were solely based on typical *Sarcina-*like morphology without genotyping. Therefore, the aim of this work was culture detection and the taxonomic classification of *Sarcina* isolates originating from different mammalian hosts. *Sarcina-*like colonies were isolated and collected during cultivation analyses of animal fecal samples (*n* = 197) from primates, dogs, calves of domestic cattle, elephants, and rhinoceroses. The study was carried out on apparently healthy animals kept in zoos or by breeders in the Czech Republic and Slovakia. Selected isolates were identified and compared using 16S rRNA gene sequencing and multi-locus sequence analysis (MLSA; *Iles, pheT, pyrG, rplB, rplC*, and *rpsC*). The results indicate the taxonomic variability of *Sarcina* isolates. *S. ventriculi* appears to be a common gut microorganism in various captive primates. In contrast, a random occurrence was also recorded in dogs. However, dog isolate N13/4e could represent the next potential novel *Sarcina* taxonomic unit. Also, a potentially novel *Sarcina* species was found in elephants, with occurrences in all tested hosts. *S. maxima* isolates were detected rarely, only in rhinoceroses. Although *Sarcina* bacteria are often linked to lethal diseases, our results indicate that *Sarcina* spp. appear to be a common member of the gut microbiota and seem to be an opportunistic pathogen. Further characterization and pathogenic analyses are required.

## 1. Introduction

The genus *Sarcina* within the *Clostridiaceae* family represents morphologically atypical, almost spherical cells forming packets, usually of eight or more units. Two taxa of *Sarcina* spp. were validly recognized. In 1842, Goodsir et al. first described *Sarcina ventriculi* in the stomach contents of a human patient with recurrent vomiting. The second recognized valid species with *Sarcina*-like morphology was *Sarcina maxima,* isolated from the feces of an elephant in 1969 [1]. *Sarcina* spp. occur naturally in soil, mud, cereal grains, and in the gastrointestinal tract of animals, as summarized by Lawson & Rainey [2]. Nevertheless, *Sarcina* findings in the digestive tract of humans and animals are often associated with various pathologies, particularly conditions related to delayed gastric emptying and food movement further into the intestine. Nausea, vomiting, ulcers, and chronic dyspepsia are other related manifestations [3,4,5,6,7]. According to a recently published review [8], more than 100 full-text articles assessing eligibility have been published. Therefore, the number of medical reports on human hosts is alarming. Similarly, in monogastric animals such as horses, dogs, and cats, gastric dilations have also been documented [9,10]. In addition, there are other cases involving gastric problems in farm animals [11,12] or chimpanzees in Sierra Leone [13] with fatal outcomes. However, some publications have described the presence of *Sarcina* spp. in healthy animals and humans [14,15,16,17].

The pathogenicity of *Sarcina* bacteria is not entirely clear and is an interesting subject for future investigation. It is striking that, until today, the identification of *Sarcina* spp. has been mainly based on its morphology and not on a molecular genetic taxonomic approach. This has been documented in several published case reports [8], in which atypical *Sarcina*-like morphology was used for *S. ventriculi* identification [7,18,19]. Only a few studies have reported proper taxonomic identification at the species level [20,21,22] compared to the highly studied and characterized clostridia. Sporadic but repeated occurrences of bacterial cells showing *Sarcina*-like morphology on selective medium for bifidobacteria [15,17] during cultivation analyses of mammalian fecal samples prompted us to target their monitoring. The goal of our study was the culture-dependent screening of *Sarcina*-like morphology colonies in fecal samples of apparently healthy animals kept in zoos or by breeders in the Czech Republic and Slovakia to get isolates for proper taxonomic identification that would highlight the variability within the genus *Sarcina* and the need for further research into this dreaded potential pathogen.

## 2. Materials and Methods

### 2.1. Ethical Statement, Sampling, and Cultivation Analysis of Fecal Samples

The sampling of animal feces was performed during routine daily procedures at zoos, farms, and breed centers (Czech Republic and Slovakia). Zoological institutions have rigorous standards for animal welfare and are accredited by the European Association of Zoos and Aquaria. All procedures involving animals adhered to the recommendations of the “Guide for the Care and Use of Animals” by the Czech University of Life Sciences Prague and to the legal requirements of the Czech Republic for the ethical treatment of nonhuman primates, as well as in accordance with European Directive 2010/63/EU. Some animal and human isolates were obtained during culture analyses of fecal samples from previously published animal studies [15,17,23].

In total, 197 fecal samples from primates (*n* = 65), dogs (*n* = 70), calves of domestic cattle (*n* = 50), elephants (*n* = 10), and rhinoceroses (*n* = 2) were collected in the years 2015–2020 with the original intention of characterizing the cultivable anaerobic bacteria. Fresh fecal samples were collected directly after host defecation. Sampling was done using a sterile spoon from the part that did not touch the ground. Then, fecal samples were placed in an anaerobically prepared medium. Media preparation, sample storage, and cultivation analysis were performed according to Modrackova et al. [17]. The repeated occurrence of the bacterium with *Sarcina*-like morphology on the modified Wilkins-Chalgren medium directed us to long-term monitoring of *Sarcina* spp. in tested fecal samples. Yellow-pigmented irregular colonies with a typical *Sarcina*-like morphology, monitored by phase contrast microscopy, were selected for future identification and characterization. Also, one human isolate of *S. maxima* 7 from a previous study [24] was included. In total, 58 *Sarcina*-like morphology isolates of different origins were selected for genotyping.

### 2.2. Colony Isolation and Identification

Putative *Sarcina* strains grown in bifidobacterial cultivation media were isolated and consecutively sub-cultivated in WSP broth under anaerobic conditions at 37 °C for 1 d [17]. Bacterial cultures were stored at room temperature (20 °C) and re-inoculated every 3 d. DNA was isolated using PrepMan Ultra^TM^ (Applied Biosystems, Waltham, MA, USA) according to the manufacturer’s instructions and stored at −20 °C. DNA samples at a concentration of 10–500 ng were used for polymerase chain reaction (PCR) amplification. Primers fd1 (5′-AGAGTTTGATCCTGGCTCAG-3′) and rP2 (5′-ACGGCTACCTTGTTACGACTT-3′) were used for PCR amplification of the 16S rRNA gene [25]. PCR products were purified using the E.Z.N.A. Cycle Pure Kit (Omega Bio-Tek, Norcross, Georgia, USA) and Sanger sequenced by Eurofins Genomics (Ebersberg, Germany). The obtained sequences were carried out in Chromas Lite 2.5.1 (Technelysium Pty Ltd., Tewantin, Australia) and BioEdit [26,27] using the ClustalW algorithm [28]. Bacteria were preliminarily classified based on a comparative analysis of the EzBioCloud-16S rRNA gene.

### 2.3. Phylogenetic Studies

The genomic DNA of 24 selected *Sarcina* isolates identified based on the 16S rRNA gene sequencing was used for multi-locus sequence analysis (MLSA). The *Iles*, *pheT*, *pyrG*, *rplB*, *rplC*, and *rpsC* operating genes encoding the isoleucyl-tRNA synthetase, phenylalanyl-tRNA synthetase beta subunit, CTP synthase, 50S ribosomal protein L2, 50S ribosomal protein L3, and 30S ribosomal protein S3, respectively, were chosen for MLSA and phylogenetic analyses. Primer pairs flanking variable sections were reconstructed using the sequences of nine bacterial strains retrieved from the complete genomes (Appendix A). The consensual sequences obtained in Geneious v7.1.7 software were employed. The obtained sequences were deposited in the NCBI database using the Banklt application (https://www.ncbi.nlm.nih.gov/WebSub/), accessed on 29 October 2022.

Phylogenetic relationships among *Sarcina* isolates, whose genomes of *Sarcina* spp. are available in the NCBI database (https://www.ncbi.nlm.nih.gov/nuccore/), accessed on 29 October 2022, and related clostridial taxa based on the 16S rRNA gene phylogeny [2] were reconstructed in the MEGA v5.05 software package using the maximum-likelihood statistical method and a particular multi-locus (ML) model [29]. Gene pairwise identities were calculated using Geneious v7.1.7 software.

### 2.4. Data Accessibility

The nucleotide sequences of the 16S rRNA gene, *Iles*, *pheT*, *pyrG*, *rplB*, *rplC*, and *rpsC* operating genes are available under the GenBank accession numbers (Appendix A).

## 3. Results

### 3.1. Occurrence of Sarcina spp. in the Feces of Screened Mammalian Hosts

Due to the recurrent occurrence of Sarcina cells on Wilkins-Chalgren agar supplemented with soya peptone, L-cysteine, Tween 80, acetic acid, and mupirocin, primarily used to isolate bifidobacteria, a total of 197 fresh fecal samples were screened for the presence of Sarcina spp. colonies. Visually distinguishable irregular yellow colonies with Sarcina-like morphology were determined by cultivation in fecal samples of 25/65 primates, 1/70 dogs, 2/50 calves of domestic cattle, 10/10 elephants, and 2/2 rhinoceroses (Table 1 and Appendix A). The frequency of Sarcina occurrence was evidently the highest in elephants, where potentially novel Sarcina species were detected even in relatively higher quantities in the Zoo Ustí nad Labem in 10^7^ CFU g^−1^ of feces than in the Zoo Liberec in 10^3^ CFU g^−1^ of feces. The presence of cultivable viable Sarcina cells in monkey feces was more common in Old World monkeys, such as guenons and gibbons, which were determined to be ordinarily from 10^5^ to 10^7^ CFU g^−1^ of fecal samples. The occurrence in dogs, calves of domestic cattle, and humans was random, and Sarcina-like cells were detected at counts of <10^4^ CFU g^−1^.

### 3.2. Identification and Taxonomic Classification of Isolated Sarcina Strains

In total, 57 mammalian Sarcina isolates from diverse animal hosts were collected, plus one human isolate and two type strains. Therefore, 60 strains were selected for genotyping (Table 2). Unfortunately, the cultivation of *Sarcina* spp. is a demanding task. This is not only because of its strictly anaerobic nature but also because of large clusters of *Sarcina-*like cell arrangements that may harbor some contaminants originating from the complex gut microbiota, which likely decreased the successful identification of a significant number of isolates. Successful 16S rRNA identification was performed for 24 of 60 DNA samples (Table 2), which were also selected for MLSA. Partial gene sequences obtained in the study using specific primers and PCR conditions (Table 3), deposited in the NCBI database with the initial letters OK (for operating genes) and MZ (for 16S rRNA), are listed in Appendix A. This table also includes the relevant genes of *Sarcina* spp. and clostridial taxa available in the NCBI database that were used to reconstruct the phylogenetic relationships.

The 16S rRNA-based phylogenetic tree (Figure 1) grouped *Sarcina* strains into four clusters, represented by strains of the species *S. ventriculi* and *S. maxima* and two other *Sarcina* sp. groups. The first included seven strains (D3/3C, K3/7B, S8/2c, S10/2a, S1/3c, K1/7A, and S2/2b) and the second strain, *Sarcina* sp. N13/4e. A very similar topology, including the four *Sarcina* clusters, was generated based on the concatenated sequences of six operating gene fragments (Figure 2) and amino acid translations (Appendix A).

Members of the *Sarcina* sp. cluster, including seven strains, have 98.62–99.08% 16S rRNA gene (1302 nucleotides) sequence similarity to type strains of *S. maxima* and *S. ventriculi*. Pairwise identities based on the *ileS* (699 nucleotides in length), *pyrG* (480), *rpsC* (444), *rplB* (516), *rplC* (360), and *pheT* (570) gene concatenates were computed in the range of 92.83–94.49%. The strain N13/4e had 98.39–90.0% 16S rRNA gene identity to *S. maxima* and *S. ventriculi* type strains and 91.92–94.04% based on the operating gene concatenate. Much higher (>98%) pairwise identities were recorded for other strains clustered with *S. maxima* and *S. ventriculi-*type strains.

Phylogenetic analyses suggested that a cluster of seven *Sarcina* strains (D3/3C, K3/7B, S8/2c, S10/2a, S1/3c, K1/7A, and S2/2b) and *Sarcina* sp. N13/4e represents a novel *Sarcina* taxonomic unit. However, sophisticated modern methods, especially those based on whole-genome sequences, must be used to confirm this assumption.

## 4. Discussion

*Sarcina* occurrences associated with the mammalian gastrointestinal microbiota were recorded in young ruminants [12,30,31], dogs [9], as well as cats [10], and primates [13,15,17,32,33,34]. Our results indicate that bacteria with *Sarcina*-like cell morphology originating from fecal samples of mammalian hosts such as primates, dogs, calves of domestic cattle, elephants, and rhinoceroses were clustered into four groups based on 16S rRNA gene sequencing and MLSA.

Animal gut microbiome seems to be significantly modified by dietary changes in the host species and geography. Dynamic microbial communities aid the living and survival of animals in changing environmental conditions, including habitat degradation, captive breeding, and diet. If this host’s gut microbial balance is disturbed and dysbiosis occurs, disease development is presumed [35,36,37,38]. Even though it was possible to isolate *Sarcina* spp. from various species of primates from different zoos in the Czech Republic and Slovakia. All these isolates were identified as *S. ventriculi* and grouped into one group. Interestingly, in Zoo Liberec (Czechia), *Sarcina*-like bacteria from primates (Northern white-cheeked gibbons) and Asian elephant hosts were isolated. However, they were identified as two different species. Based on our results, the taxonomic variability of the analyzed *Sarcina* strains is more dependent on the animal species than the host location. This confirms the fact that isolates from elephants from two different zoos were also clustered into one group (potentially novel *Sarcina* sp.). *S. ventriculi* seems to be a common gut microorganism in captive primates such as guenons (De Brazza’s monkey, Hamlyn’s monkey, Roloway monkey, Lesser spot-nosed monkey, and Vervet monkey) and gibbons (northern white-cheeked gibbon, gibbon siamang, and Yellow-cheeked crested gibbon). *S. ventriculi* was also detected in the feces of ring-tailed lemurs and golden lion tamarins. Samples of primates belonging to New World monkeys, such as marmosets and other tamarins, were also subjected to culture analysis under the same conditions; however, *Sarcina*-like cells were detected rarely. Interestingly, *S. ventriculi* was detected only in one of the eight individuals belonging to the golden lion tamarin species. Similarly, other authors also described *S. ventriculi* presence in the feces of Old Word monkeys such as guenons and crested gibbons [15,22], macaques [32], wild gorillas [33], and chimpanzees [13,34]. Interestingly, these records indicate that *S. ventriculi* seems to be common not only for animals kept in captivity but also for primates living in the wild or on nature reserves. The more frequent and successful detection of *S. ventriculi* may be influenced by the lower representation of bifidobacteria in the microbiota of Old World monkeys compared to New World monkeys [17], which allows the growth of distinguishable *Sarcina*-like bacterial colonies and their possible isolation and identification. Also, the fecal microbiota of the dog is known for the relatively high occurrence of viable clostridial cells and low or culture-undetectable numbers of bifidobacteria in dog feces which then allow detection of the *Clostridiaceae* family on bifidobacterial media [23].

*Sarcina* species have an atypical morphology, with almost spherical cells forming packets, usually of eight or more cells, which is rarely observed in other bacteria. The occurrence of visually distinguishable irregular yellow colonies with *Sarcina*-like morphology on modified Wilkins-Chalgren agar during the cultivation of primate fecal samples has been previously described [15,17]. Wilkins-Chalgren agar is used to cultivate anaerobic bacteria. Media modification with soya peptone, L-cysteine, Tween 80, acetic acid, and mupirocin is primarily used to isolate *Bifidobacterium* species [39]. S. ventriculi and S. *maxima* are anaerobes known for their ability to grow at very low pH values [1,40], and resistance to mupirocin has also been detected [15]. Therefore, the selective factors used in modified Wilkins-Chalgren agar do not limit the cultivation of viable *Sarcina* species present in the tested fecal samples, and the media used seem to be suitable for their cultivation and detection as well.

A potentially novel *Sarcina* species represented by seven strains (D3/3C, K3/7B, S8/2c, S10/2a, S1/3c, K1/7A, and S2/2b) isolated from different Asian elephant hosts from the Zoo Liberec and the Zoo Ústí nad Labem was found to be a common commensal of elephants. It is a paradox that *S. maxina* as a taxon was first described in 1969 in an elephant [1]. However, we identified *S. maxima* isolates in a human vegetarian host and in both rhinoceroses. Only one *Sarcina* sp. occurrence was recorded on the modified Wilkins-Chalgren medium out of the 70 analyzed dog fecal samples. Interestingly, this isolate could represent the next novel *Sarcina* taxonomic unit.

A recently published study [13] dealt with the occurrence of *Sarcina* bacteria in connection with chimpanzee deaths in Sierra Leone, which were further isolated and characterized. By studying the morphology and growth characteristics of these strains and by completing genome sequencing of an isolate, researchers identified features that distinguish “*Candidatus Sarcina troglodytae*” from all previously described members of the *Sarcina* genus. They showed that the bacterium possesses genes encoding biochemical pathways that potentially contribute to enhanced virulence, including an encoded urea degradation biochemical pathway, consistent with the clinical signs observed in chimpanzees. Overall, they concluded that the *Sarcina* genus probably comprises an overlooked complex of species, ranging from benign commensals to frank pathogens. However, further analyses leading to an understanding of its pathogenicity are often lacking. In addition, most published cases [8] involved the identification of *Sarcina* isolates only at the genus level based on the morphology and health complications of the host. The threshold value for distinguishing between the two species was 98.65% [41]. Our results were based on 16S rRNA gene sequencing and MLSA (*Iles*, *pheT*, *pyrG*, *rplB*, *rplC*, and *rpsC* genes), indicating species variability across different hosts. In addition to the two known recognized taxa (*S. ventriculi* and *S. maxima*), two potentially novel species isolated from dogs and elephants were considered. Based on 16S rRNA gene sequencing and MLSA, there is an assumption of *Sarcina* variability at the genome level. Remarkably, these analyses clustered the chimpanzee isolates “*Candidatus Sarcina troglodytae*” [13] into the *S. ventriculi* taxon together with other primates isolates. Therefore, future complete genome sequencing of our *Sarcina* isolates can bring new points of interest about their properties.

## 5. Conclusions

Our screening results indicate the common presence of *Sarcina* spp. in the gut microbiota of various mammals kept in captivity without obvious health complications. *Sarcina* spp. occurrence was more significantly affected at the level of animal species than at the level of host location. *S. ventriculi* appears to be a common part of the primate intestinal microbiota, especially those of Old World monkeys. The presence of potentially novel taxa across detected mammals can bring another view on *Sarcina* function in the gut microbiome of animals.

## Figures and Tables

**Figure 1 animals-13-01529-f001:**
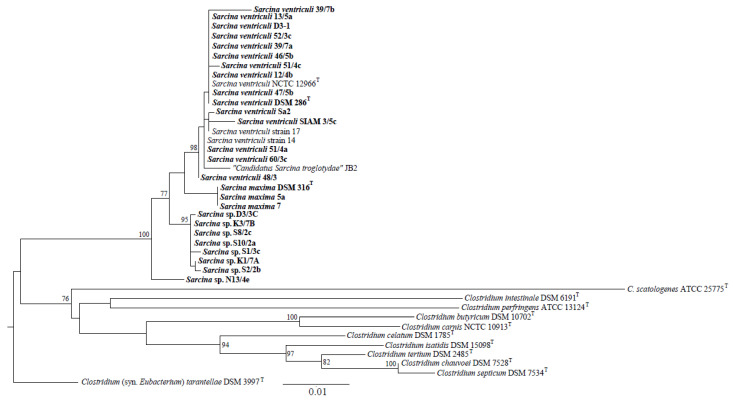
Phylogenetic relationships among *Sarcina* and related *Clostridium* strains based on 16S rRNA gene sequences (1302 nucleotides in length). The Kimura 2 + G + I best-fit multi-locus (ML) evolutionary model was applied for the reconstruction. The phylogeny was supported by bootstrapping (1000 datasets), while values > 70% are shown at particular nodes. Scale: 0.01 substitutions per nucleotide position. *Sarcina* isolates whose DNA was used for 16S rRNA sequencing are marked in bold.

**Figure 2 animals-13-01529-f002:**
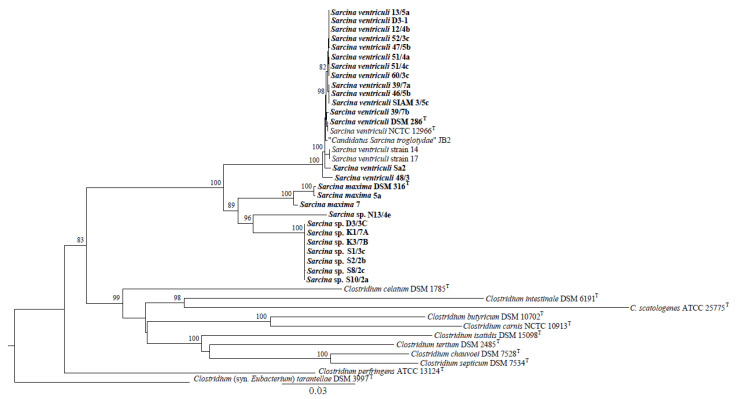
Maximum likelihood phylogenetic reconstruction reveals relationships among *Sarcina* strains based on a concatenate of partial *ileS* (699 nucleotides in length), *pyrG* (480), *rpsC* (444), *rplB* (516), *rplC* (360), and *pheT* (570) gene sequences. The GTR + G + I best-fit multi-locus (ML) model and bootstrap values > 70% (from 1000 replicates) at particular nodes were applied for tree modeling. Bar, 0.03 substitutions per nucleotide position. *Sarcina* isolates whose DNA was used for multi-locus sequence analysis (MLSA) sequencing are marked in bold.

**Table 1 animals-13-01529-t001:** List of animal hosts (*n* = 197) whose feces were subjected to culture screening for *Sarcina* spp. (number of positive samples/number of analyzed fecal samples; animal species with detected *Sarcina*-like isolates are marked in bold).

**PRIMATES (*n* = 65)**	Moustached tamarin (0/2)	Cavalier King Charles spaniel (0/1)
Brown-mantled tamarin (0/1)	**Northern Talapoin monkey (1/1)**	Crossbreed dog (0/12)
**Campbell’s Mona monkey (1/1)**	**Northern white-cheeked gibbon (3/4)**	Czechoslovakian wolfdog (0/2)
**Common marmoset (1/3)**	**Patas monkey (2/3)**	Foxterier (0/1)
Cotton-top tamarin (0/1)	**Putty-nosed monkey (1/1)**	German Shepherd dog (0/28)
**De Brazza’s monkey (1/1)**	Pygmy marmoset (0/1)	Golden retriever (0/2)
Diana monkey (0/1)	Red-handed tamarin (0/2)	Havanese (0/3)
Emperor tamarin (0/4)	**Ring-tailed lemur (4/4)**	Labrador retriever (0/1)
**Gibbon siamang (3/3)**	**Roloway monkey (1/1)**	Not known (0/1)
Goeldi’s marmoset (0/1)	Silvery marmoset (0/4)	Samoyed (0/1)
**Golden lion tamarin (1/8)**	**Yellow-cheeked crested gibbon (3/4)**	Swiss shepherd (0/2)
Golden-bellied mangabey (0/1)	White-faced saki (0/1)	Whippet (0/1)
**Vervet monkey (1/1)**	White-headed marmoset (0/2)	**OTHERS (*n* = 62)**
Hamadryas baboon (0/1)	**DOGS (*n* = 70)**	**Asian elephant (10/10)**
**Hamlyn’s monkey (1/1)**	**American staffordshire terrier (1/1)**	**Eastern black rhinoceros (2/2)**
Chimpanzee (0/5)	Basenji (0/1)	**Holstein-Friesian calf (2/50)**
**Lesser spot-nosed monkey (1/1)**	Belgian shepherd (0/2)	
Lion-tailed macaque (0/1)	Border collie (0/11)	

**Table 2 animals-13-01529-t002:** List of isolates with *Sarcina*-like morphology, including origin and location of the hosts and species identity based on comparative analysis of the EzBioCloud-16S rRNA gene (selected *Sarcina* spp. for genotyping are marked in bold).

Isolated from		Host Location	*Sarcina* Isolate	16S rRNA Identification	Similarity (%)	nts
Human (vegetarian)	*Homo sapiens*	Prague, CZ	**7**	*Sarcina maxima*	99.63	1356
Northern white-cheeked gibbon	*Nomascus leucogenys*	Zoo Liberec, CZ	40/5a	NRI		
			40/5b	NRI		
			59/5a	NRI		
			60/2	NRI		
			60/3a	NRI		
			**60/3c**	*Sarcina ventriculi*	99.58	1419
Gibon siamang	*Symphalangus syndactylus*	Zoo Olomouc, CZ	SIAM 3/5b	*Sarcina ventriculi*	99.33	1347
			**SIAM 3/5c**	*Sarcina ventriculi*	99.41	1362
Yellow-cheeked crested gibbon	*Nomascus gabriellae*	Zoo Bratislava, SK	**51/4a**	*Sarcina ventriculi*	99.50	1414
			51/4b	NRI		
			**51/4c**	*Sarcina ventriculi*	99.01	1422
Yellow-cheeked crested gibbon	*Nomascus gabriellae*	Zoo Olomouc, CZ	46/4a	NRI		
			46/4c	NRI		
			**46/5b**	*Sarcina ventriculi*	99.64	1420
			46/6	NRI		
			47/4b	NRI		
			**47/5b**	*Sarcina ventriculi*	99.50	1414
De Brazza’s monkey	*Cercopithecus neglectus*	Zoo Plzeň, CZ	39/5a	NRI		
			39/6a	NRI		
			**39/7a**	*Sarcina ventriculi*	99.56	1367
			**39/7b**	*Sarcina ventriculi/Sarcina* sp.	98.86	1410
Hamlyn’s monkey	*Cercopithecus hamlyni*	Zoo Bojnice, SK	55/5a	NRI		
Roloway monkey	*Cercopithecus roloway*	Zoo Bojnice, SK	56/4a	NRI		
			56/4b	NRI		
			56/5	NRI		
Lesser spot-nosed monkey	*Cercopithecus petaurista*	Zoo Bojnice, SK	57/3	NRI		
			57/4	NRI		
			57/4a	NRI		
			57/4b	NRI		
			57/4c	NRI		
			57/5	NRI		
			57/5a	NRI		
			57/5b	NRI		
Vervet monkey	*Chlorocebus sabaeus*	Zoo Hodonín, CZ	52/3a	*Sarcina ventriculi*	99.50	1397
			52/3b	*Sarcina ventriculi*	99.56	1352
			**52/3c**	*Sarcina ventriculi*	99.58	1424
Ring-tailed lemur	*Lemur catta*	Zoo Olomouc, CZ	**12/4b**	*Sarcina ventriculi*	99.56	1382
			**13/5a**	*Sarcina ventriculi*	99.56	1371
			**48/3**	*Sarcina ventriculi*	99.14	1406
Golden lion tamarin	*Leontopithecus rosalia*	Zoo Olomouc, CZ	**D3-1**	*Sarcina ventriculi*	99.63	1355
Eastern black rhinoceros	*Diceros bicornis michaeli*	Safari park Dvůr Králové, CZ	**5a**	*Sarcina maxima*	100	1403
			5b	*Sarcina maxima*	99.92	1308
Asian elephant	*Elephas maximus*	Zoo Liberec, CZ	S1/2a	*Sarcina maxima*/*Sarcina* sp.	98.89	1352
			S1/2c	NRI		
			S1/3b	NRI		
			**S1/3c**	*Sarcina maxima*/*Sarcina* sp.	98.74	1354
			S1/3d	NRI		
			**S10/2a**	*Sarcina maxima*/*Sarcina* sp.	98.88	1346
			**S2/2b**	*Sarcina maxima*/*Sarcina* sp.	98.83	1371
			S4/2c	NRI		
			**S8/2c**	*Sarcina ventriculi*/*Sarcina* sp.	98.82	1356
Asian elephant	*Elephas maximus*	Zoo Ústí nad Labem, CZ	K1/6A	NRI		
			**K1/7A**	*Sarcina ventriculi*	98.95	1241
			**K3/7B**	*Sarcina maxima*	98.74	1356
			D1/5A	NRI		
			**D3/3C**	*Sarcina maxima*	99.06	1273
Holstein-Friesian calf	*Bos taurus*	Vražkov, CZ	**Sa1**	NRI		
			**Sa2**	*Sarcina ventriculi*	99.56	1370
Dog	*Canis lupus* f. *familiaris*	Prague, CZ	**N13/4e**	*Sarcina maxima*/*Sarcina* sp.	98.90	1371
Soil		DSMZ—type strain	**DSM 286^T^**	*Sarcina ventriculi*	99.57	1392
Elephant		DSMZ—type strain	**DSM 316^T^**	*Sarcina maxima*	100	1349

Footnotes: nts, obtained number of nucleotides for the EzBioCloud-16S rRNA gene comparative analysis; NRI, not reliable identification (probably due to hidden contamination); CZ, Czech Republic; SK, Slovakia.

**Table 3 animals-13-01529-t003:** Primer sequences and PCR conditions for amplification of particular operating gene fragments in *Sarcina*-related bacteria.

Gene	Forward/Reverse Primers (5′ → 3′) PCR Conditions	Length of the Amplified/Sequenced Fragment (Nucleotides)
*ileS*	ileSSarF: TGGACAACAACTCCGTGG/ileSSarR: TGACCACATTCACATTCCC	699
	95 °C for 5 min; 30 × (95 °C for 45 s; 56 °C for 40 s; 72 °C for 50 s); 72 °C for 6 min	
*pheT*	pheTSarF: TGTAACGGAAAGAGAGCC/pheTSarR: ATCTAAATCAAGCTCTGCC	570
	95 °C for 5 min; 30 × (95 °C for 45 s; 53 °C for 40 s; 72 °C for 50 s); 72 °C for 6 min	
*pyrG*	PyrGSarF: ACAGCAGCATCTTTAGG/PyrGSarR: AACTCTCCTGCTTTTCC	480
	95 °C for 5 min; 30 × (95 °C for 45 s; 53 °C for 40 s; 72 °C for 50 s); 72 °C for 6 min	
*rplB*	rplBSarF: GGTGGTAGAAATGGTCAAGG/rplBSarR: ATCCTCTAACAGTAGGTCTGA	516
	95 °C for 5 min; 30 × (95 °C for 45 s; 56 °C for 40 s; 72 °C for 50 s); 72 °C for 6 min	
*rplC*	rplCSarF: CCAGTAACAGTTGTAGAAGC/rplCSarR: CTTGATGGATCTGATGAAGC	360
	95 °C for 5 min; 30 × (95 °C for 45 s; 53 °C for 40 s; 72 °C for 50 s); 72 °C for 6 min	
*rpsC*	rpsCSarF: CTCACGGACTAAGAGTTGG/rpsCSarR: TTTCTGCACCACCTAATCTACC	444
	95 °C for 5 min; 30 × (95 °C for 45 s; 56 °C for 40 s; 72 °C for 50 s); 72 °C for 6 min	

## Data Availability

The nucleotide sequences of the 16S rRNA gene, *Iles*, *pheT*, *pyrG*, *rplB*, *rplC*, and *rpsC* operating genes are available under the GenBank accession numbers presented in Appendix A, and other data are available on request from the corresponding author.

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
