# Peer review of "Species and Strain Variability among *Sarcina* Isolates from Diverse Mammalian Hosts"

_animals, 2023, doi:10.3390/ani13091529_

Round 1
Reviewer 1 Report
The manuscript “Animals-2288729” aims to describe the frequency of occurrence, identification and subtyping of Sarcina isolates from fecal samples in diverse mammalian hosts from Czech Republic. The study is overall well written, and I believe it is of interest to its field. However, the objectives of the study are not clearly stated, and the bounds of the study are not properly described. For example, it is not clear if the animals from the study were apparently healthy or had any disease, but the pathogenic potential of Sarcina bacteria is mentioned in both the introduction and the discussion. In the discussion there should also be a section on how captivity (zoo specimens) may have affected the results of the study (frequency of detection and diversity by host), and a discussion about the phylogenetic relationships of the isolates by spatial clustering and hosts.
Line 23: what are “cases” without apparent host health problems? Cases of what? This sentence is confusing, I suggest rewriting together with the previous one.
Line 23-24: the abstract starts stating that Sarcina ventriculi has been detected associated with clinical conditions and apparently healthy hosts, but in this sentence the authors claim that in most of these articles the identification was not performed. Do the authors refer to strain subtyping of Sarcina ventriculi? Please rewrite to clarify this sentence.
Line 24-25: what is culture determination?
Line 26: “Dozens” is not a precise term, please correct.
Line 26: Sarcina-like colonies
Line 26-27: Please indicate the bounds of the study, state if the study was performed on apparently healthy hosts or diseased, the location where the study was performed, etc…
Line 43: I believe it is more accuarate to refer as “Sarcina-like morphology”, please correct throughout the article.
Line 63-68: The objectives of the study are not clearly defined. Please state the objectives more clearly, including the bounds of the study (e.g. the study is performed in Czech Republic, not the whole world). Are the samples originating from apparently healthy animals? Were the animals selected because of a health condition?
Line 55: “Candidatus Sarcina troglodytae”, all scientific names should be italicised.
Line 64: if “sarcina” refers to a bacteria genus should be in capitals and italics. Please review the whole document.
Line 65: it is unclear what the authors are refereeing as “her”. If this pronoun refers to Sarcina please substitute by “its” as gender does not apply for bacteria.
Line 72: Table S2 should be mentioned in this section. Scientific names should be included in Table S2, and I also suggest adding the location and year of each isolate in the table.
Line 92: I suggest to the authors to remove “future” as it introduces confusion and does not add any sense in the sentence. The same occurs in other sections, please review (e.g. line 111).
Line 132: it is unclear if fecal samples belong to different individuals or perhaps some samples analyzed belong to the same individual. This is of relevance for the frequencies provided. Please clarify in the text.
Line 132: Perhaps the authors could include a figure to visualize the frequency of detection of Sarcina by host/breed species, this can help the reader without the need to check the details in Table S2.
Line 145-146: I suggest removing this sentence and move Table 1 to the next section about the subsequent identification and subtyping.
Line 148: Please complete the table title for a comprehensive description of the content of the table (e.g. in this table there is also information on identification by the 16S rRNA). Please describe all abbreviations that appear in the table (e.g. “nts”)
Line 161: I suggest moving table 3 into supplementary materials, this is a huge table that does not necessarily helps to understand the article. Instead, I suggest marking the strains studied in this work in Figure 1.
Line 185-186: please rewrite the title of table 3 title as it is not understandable.
Line 205: please correct, “in captive primates”.
Author Response
Reviewer 1
The he manuscript “Animals-2288729” aims to describe the frequency of occurrence, identification and subtyping of Sarcina isolates from fecal samples in diverse mammalian hosts from Czech Republic. The study is overall well written, and I believe it is of interest to its field. However, the objectives of the study are not clearly stated, and the bounds of the study are not properly described. For example, it is not clear if the animals from the study were apparently healthy or had any disease, but the pathogenic potential of Sarcina bacteria is mentioned in both the introduction and the discussion. In the discussion there should also be a section on how captivity (zoo specimens) may have affected the results of the study (frequency of detection and diversity by host), and a discussion about the phylogenetic relationships of the isolates by spatial clustering and hosts.
- We tried to modify the manuscript as recommended above. The introduction was rewritten to be more targeted, and the objectives of the study were also better specified. Part of the results was included in the discussion, and everything was more connected and discussed. The summary has also been revised. All changes are marked in yellow in the revised version of the manuscript. Due to many changes, it was not possible to use tracking changes; the document would be unreadable.
Line 23: what are “cases” without apparent host health problems? Cases of what? This sentence is confusing, I suggest rewriting together with the previous one.
- We changed it to “medical case reports” (lines 24-25; 30-31).
Line 23-24: the abstract starts stating that Sarcina ventriculi has been detected associated with clinical conditions and apparently healthy hosts, but in this sentence the authors claim that in most of these articles the identification was not performed. Do the authors refer to strain subtyping of Sarcina ventriculi? Please rewrite to clarify this sentence.
- See the lines 23-31, we updated.
Line 24-25: what is culture determination?
- Corrected to “culture detection” line 27.
Line 26: “Dozens” is not a precise term, please correct.
- It was deleted and the sentence corrected (lines 28-30).
Line 26: Sarcina-like colonies
- It was corrected though the manuscript (line 26, 28, 51, 67, 71, 95, 97, 100). All is marked by yellow color.
Line 26-27: Please indicate the bounds of the study, state if the study was performed on apparently healthy hosts or diseased, the location where the study was performed, etc…
- We indicated (lines 30-31, 72-73).
Line 43: I believe it is more accuarate to refer as “Sarcina-like morphology”, please correct throughout the article.
- It was corrected though the manuscript as specified above.
Line 63-68: The objectives of the study are not clearly defined. Please state the objectives more clearly, including the bounds of the study (e.g. the study is performed in Czech Republic, not the whole world). Are the samples originating from apparently healthy animals? Were the animals selected because of a health condition?
- We indicated (lines 30-31, 72-73, 85-90).
Line 55: “Candidatus Sarcina troglodytae”, all scientific names should be italicised.
- It was corrected (lines 273, 288, and in Fig. 1, 2, S1).
Line 64: if “sarcina” refers to a bacteria genus should be in capitals and italics. Please review the whole document.
- As I described above, we change often to “Sarcina-like”morphology, cells, or colonies, or for Sarcina xxx, as is requested.
Line 65: it is unclear what the authors are refereeing as “her”. If this pronoun refers to Sarcina please substitute by “its” as gender does not apply for bacteria.
- The “her” was deleted, and the sentence changed (lines 70-73).
Line 72: Table S2 should be mentioned in this section. Scientific names should be included in Table S2, and I also suggest adding the location and year of each isolate in the table.
- The scientific names are in Table 2, which is a list of Sarcina spp. isolates. Origin of the samples; host (English and Latin), and host location are included. Date of isolation, etc. are data connected to NCBI number (Table S3).
Line 92: I suggest to the authors to remove “future” as it introduces confusion and does not add any sense in the sentence. The same occurs in other sections, please review (e.g. line 111).
- It was changed for “genotyping” (line 100), we reduced the used of “future” trough the manuscript.
Line 132: it is unclear if fecal samples belong to different individuals or perhaps some samples analyzed belong to the same individual. This is of relevance for the frequencies provided. Please clarify in the text.
- It was changed (line 140).
Line 132: Perhaps the authors could include a figure to visualize the frequency of detection of Sarcina by host/breed species, this can help the reader without the need to check the details in Table S2.
- We added to the main paper Table 1 (where is the list of analyzed animals), but we were not able to find a form for figure visualization. The data are connected to Table S2, and the footnotes are references for detailed information.
Line 145-146: I suggest removing this sentence and move Table 1 to the next section about the subsequent identification and subtyping.
- The sentence was moved (lines 162-164).
Line 148: Please complete the table title for a comprehensive description of the content of the table (e.g. in this table there is also information on identification by the 16S rRNA). Please describe all abbreviations that appear in the table (e.g. “nts”)
- Currently, it is Table 2. The title was updated (lines 177-179), the content of the table was better formatted, and some data added to Footnotes (lines 180-181).
Line 161: I suggest moving table 3 into supplementary materials, this is a huge table that does not necessarily helps to understand the article. Instead, I suggest marking the strains studied in this work in Figure 1.
- The Table 3 was moved into supplementary materials, currently is marked as Table S3. The strains whose were used were marked in bolt and Fig. 1, 2, S1.
Line 185-186: please rewrite the title of table 3 title as it is not understandable.
- The title was updated (Suppl. Materials).
Line 205: please correct, “in captive primates”.
- It was corrected (line 233-234), the recommendation was used also in abstract (line 34).
Reviewer 2 Report
The article deals with an interesting topic and was well written with clarity of objectives, methods, results.
The considerations are highlighted in the text.

Author Response
We tried to modify the manuscript as recommended in commented version. All changes are marked in yellow in the revised version of the manuscript. Due to many changes, it was not possible to use tracking changes; the document would be unreadable.
- The end part of the simple summary was better to discuss. Also, in the abstract is many changes, especially in the first half.
- The introduction was rewritten to be more targeted (we used some modified sentences from the discussion, which was recommended).
- Fecal sampling was described in more detail as requested (lines 90-94).
- Table 3 was moved to supplementary materials Table S3.
- Part of the results (3.3) was included in the discussion, and everything was more connected and discussed. The conclusion has also been revised. We also added a recommended article (pH) and discussed Sarcina/Clostridiaceae occurrence in dogs, with a connection to bifidobacterial media.
- The origin of isolates is not part of phylogenetics trees; therefore, we did not add this data. However, the isolates used in our analyses are highlighted/marked.
Reviewer 3 Report
In the study animals-2288729 entitled “Species and strain variability among Sarcina isolates from diverse mammalian hosts” the authors investigate the culture determination and the taxonomic identification of Sarcina spp isolates originating from different mammalian hosts. The manuscript presents interesting new data, and that the topic of the manuscript is of significant interest and appropriate for the Journal. The manuscript is written in an acceptable English language. The presentation and the length of the manuscript are adequate as the description of the experimental plan. In particular, the title accurately reflects the major findings of the work; the keywords represent the article adequately and the abstract section well summarizes the background, methodology, results, and significance of the study; the introduction section is well written, the topic of the study is well stated and supported by adequate bibliographic information; material and methods section is well written, adequately and meticulously describes the methods applied in the study; results section is clear and the obtained findings were well explained. The discussion section is not clear and the conclusions are not well supported by the results, please rewrote these sections. However, some major changes I suggest throughout the text. In view of this, I believe that the manuscript is suitable for publication after MAJOR REVISION.
SPECIFIC COMMENTS:
The manuscript presents many typing mistakes, please correct these. The title accurately reflects the major findings of the work. The abstract clearly summarize the background, methodology, results, and significance of the study. However, AA should improve some sentences. Please rewrite and consolidate the study results, rewrite discussion and conclusion of the study. Tables and figures are generally good, and they well represent results obtained. Data in Tables were not duplicated in the text. The reference list should be improved. Correct and uniform the references during the text to the journal style.
Author Response
We tried to modify the manuscript as recommended above. The introduction was rewritten to be more targeted, and the study's objectives were also better specified. Part of the results was included in the discussion, and everything was more connected and discussed. The conclusion has also been revised. All changes are marked in yellow in the revised version of the manuscript. Due to many changes, it was not possible to use tracking changes; the document would be unreadable.
Also, the abstract was updated, as you can see. We tried to reduce typing mistakes, do better organization of Figs and Table in the manuscript, and English was discussed with native speakers and some terms with a zoologist.
Round 2
Reviewer 1 Report
The authors have addressed most of my comments in this reviewed version of the manuscript animals-2288729. I have few specific comments about this version of the manuscript:
Line 67-68: this sentence has not much sense without a context. Please provide the context in which you were finding sporadic but repeated occurences of Sarcina-like bacteria.
Line 69: Please mention first your objectives before describing information on your methods. However, this is not the section where the methods should be described, instead, you should include the information of “apparently healthy animals” in your objectives as this is part of the bounds of your study. Your objectives target apparently healthy mammalian hosts in Czech Republic and Slovakia, and these concepts should be clearly stated in your objectives (you cannot conclude on what is happening in other regions or areas or about the detection of Sarcina in diseased animals). Please review carefully this section, and make sure your objectives are consistent with your methods.
Line 76: according to modifications in the abstract and a table, the study was also performed in centres from Slovakia? Please correct or explain it, review it throughout your manuscript.
Line 154: I do not recommend starting a section (3.2.) with “Finally”, plus does not add any relevant information.
Line 155: were analyzed for what? Please complete the sentence.
Line 211: please include a section to discuss how captivity may have affected the detection of Sarcina in wildlife specimens and if there is any spatial relationship of the strains and the location of the sample (e.g. cluster of strains by zoo or farm or geographic origin). This is a relevant information when discussing about relationships among isolates.
Line 278-280: I strongly recommend to the authors not starting the conclusions of the study stating what they think it is obvious from other studies. Please rewrite your conclusions carefully, in the study you describe other relevant findings that are not included in the conclusions section.
- Table 1 have a weird format in the pdf file I have received, the columns are not clear. Please make sure that this is corrected in the final version.
Author Response
We thank the reviewer for the next helpful comments. We hope, that the current revision will be suitable for the acceptance.
All changes are by "tracking changes" and are commented on, see below (mentioned lines).
The authors have addressed most of my comments in this reviewed version of the manuscript animals-2288729. I have few specific comments about this version of the manuscript:
Line 67-68: this sentence has not much sense without a context. Please provide the context in which you were finding sporadic but repeated occurences of Sarcina-like bacteria.
- The sentence was changed (lines 67-70). Hopefully, the context should be clearer.
Line 69: Please mention first your objectives before describing information on your methods. However, this is not the section where the methods should be described, instead, you should include the information of “apparently healthy animals” in your objectives as this is part of the bounds of your study. Your objectives target apparently healthy mammalian hosts in Czech Republic and Slovakia, and these concepts should be clearly stated in your objectives (you cannot conclude on what is happening in other regions or areas or about the detection of Sarcina in diseased animals). Please review carefully this section, and make sure your objectives are consistent with your methods.
- We better connected this part with the sentence mentioned above. We added the requested, and the goal was rewritten (lines 71-75).
Line 76: according to modifications in the abstract and a table, the study was also performed in centres from Slovakia? Please correct or explain it, review it throughout your manuscript.
- We added Slovakia (line 79) and checked in the manuscript.
Line 154: I do not recommend starting a section (3.2.) with “Finally”, plus does not add any relevant information.
- „Finally“ was deleted and sentence changed (lines 158-160).
Line 155: were analyzed for what? Please complete the sentence.
- The sentence was changed and information added (lines 158-160).
Line 211: please include a section to discuss how captivity may have affected the detection of Sarcina in wildlife specimens and if there is any spatial relationship of the strains and the location of the sample (e.g. cluster of strains by zoo or farm or geographic origin). This is a relevant information when discussing about relationships among isolates.
We discussed, what is requested above. Therefore, the discussion organization was changed, and 4 new references were added (now 35-38); see lines (214-239, 248-252, plus changes in the order of references).
Line 278-280: I strongly recommend to the authors not starting the conclusions of the study stating what they think it is obvious from other studies. Please rewrite your conclusions carefully, in the study you describe other relevant findings that are not included in the conclusions section.
- It is true, we cannot present our stating, which is obvious from previous studies. The conclusion was changed (lines 303-312).
Table 1 have a weird format in the pdf file I have received, the columns are not clear. Please make sure that this is corrected in the final version.
- The Table was formatted like is correct. We will checked again in final version of the manuscript.
Reviewer 3 Report
The manuscript is now suitable for publication.
Author Response
There are no other requirements. Therefore, we consider this review to be closed. Thanks for the previous helpful advice and comments.